# Uncertainty Quantification Revisited: Conditional Cross-Entropy for Neural Classifiers

## Abstract

We propose correctness-aligned uncertainty (CAU) scores, a new family of uncertainty quantification (UQ) measures that explicitly account for whether predictions are correct or incorrect. Unlike conventional UQ, which relies on a single averaged score that conflates confidence across outcomes, CAU separates uncertainty contributions by correctness, providing richer interpretability and actionable insights. Our approach leverages conditional cross-entropy (CCE) to quantify the dissimilarity between estimated and target uncertainty distributions, yielding two complementary CAU metrics for correct and incorrect decisions. This correctness alignment naturally bridges multi-class UQ with binary decision theory, enabling principled formulations grounded in Neyman–Pearson and Bayesian detection principles. Beyond interpretability, CAU scores serve as effective tools for model refinement, misclassification detection, and performance improvement. We further highlight challenges in neural classifiers, where class imbalance between correct and incorrect predictions causes conventional UQ-based formulations to collapse to accuracy-dominated measures, thereby exposing the limitations of standard UQ approaches. We empirically demonstrate that CAU maintains robust uncertainty assessment for models with varying accuracy levels, where existing metrics often fail. Moreover, we show that incorporating CAU into the training objective improves the separation of uncertainty between correct and incorrect predictions, thereby enhancing post-processing misclassification detection.

## 1 Introduction

In high-stake applications such as medical diagnosis, autonomous driving, and financial decision-making, it is desired that deep neural networks (DNNs) not only provide accurate predictions but also deliver confidence guarantees about their predictions, to facilitate practitioners knowing when a model is likely to be correct and, crucially, when it might be making errors. To address the extended requirement, uncertainty quantification (UQ) provides principled methods to assess model confidence and identify potentially erroneous predictions (He et al., 2025).

To quantify uncertainty of machine learning models, variance-based decomposition and propagation methods are widely studied thanks to their theoretical guarantees and interpretability (Geman et al., 1992). These methods attribute prediction uncertainty to the variance of various random error sources and have spawned multiple mature implementation techniques. Among them, bootstrap sampling (Efron, 1992) or ensemble learning (Breiman, 1996) based empirical estimation methods, as well as Bayesian grounded posterior inference methods (MacKay, 1992; Neal, 1996), have been extensively validated across various traditional regression models. In classification, however, the "variance" term loses its intuitive meaning and encounters significant definitional difficulties, which hinders the direct application of variance analysis to classifiers (Domingos, 2000). These challenges become even more pronounced in DNNs, where high nonlinearity, massive parameter scales, and complex hierarchical structures make traditional analytical variance propagation analysis computationally intractable (Goodfellow et al., 2016).

To address the reliability requirement for neural classifiers, Conformal Prediction provides distribution-free statistical guarantees by constructing prediction sets that contain the true label with a specified probability (Angelopoulos & Bates, 2023). Calibration methods such as temperature scaling adjust model outputs to align predicted probabilities with actual correctness rates (Guo

et al., 2017). Evidential Deep Learning represents a direct UQ approach that models evidence for predictions using subjective logic (Sensoy et al., 2018). However, these methods either fail to produce direct uncertainty estimates or lack strong theoretical guarantees. On the contrary, self-entropy (SE) based UQ methods that directly compute the Shannon entropy of the softmax output enjoy both solid theoretical foundation and computational efficiency, however, it often produces overly confident outputs due to its relationship with cross-entropy training loss (Guo et al., 2017). To address this overconfidence issue and obtain more robust and precise probability outputs, methods such as Monte Carlo Dropout (MC-Dropout) (Gal & Ghahramani, 2016) and deep ensembles (Lakshminarayanan et al., 2017) are employed as numerical acquisition techniques for self-entropy computation. Both approaches provide more reliable probability estimates through approximate Bayesian inference, which can then be directly used for self-entropy calculation. While these techniques improve the quality of entropy estimation by mitigating overconfidence, they do not fundamentally change the core UQ principle, i.e., the Shannon entropy. Further, as a single compact score, particularly in its empirical average form, SE lacks adequate information to reliably compare uncertainty across systems on the same task.

Empirically, incorrect predictions typically exhibit higher entropy outputs compared to correct predictions (Shamsi et al., 2023; Zhang et al., 2023). With this principle, effective cross-system comparison should intuitively capture how well different models separate probability density between correct and incorrect predictions, distinguishing between well-calibrated models that appropriately express uncertainty values and overconfident models that provide low values regardless of correctness. Align with the separation principle, two primary evaluation approaches are usually adopted to compare performance across systems. The first approach utilize moment-based measures and relies on the mean and variance of uncertainty scores to distinguish correct and incorrect predictions, exemplified by Zhang et al. (2023). However, these low-order statistical moments are too coarse and insufficient to capture subtle distributional differences in model outputs. Moreover, the resulting multi-dimensional statistics lack clear priority ordering for determining which model exhibits superior uncertainty performance. The second approach extends post-classification metrics such as AUROC for misclassification detection to evaluate the quality of UQ systems. This approach originated from Hendrycks & Gimpel (2017) and has been extensively utilized thanks to its theoretical grounding in confidence calibration principles: well-calibrated models should exhibit higher confidence for correct predictions, naturally extending to the uncertainty setting (Liang et al., 2018; Liu et al., 2020; He et al., 2025; Guo et al., 2017). However, since it relies on sampling-based evaluation, thus primarily captures frequency-based statistics rather than focusing on probability density differences. Given that practical neural classifiers inevitably produce far more correct predictions than incorrect ones, and frequency-based statistics are sensitive to input sample proportions, post-classification metrics severely suffer from data imbalance issues that can mask the true separation quality between correct and incorrect predictions.

To overcome these limitations in current practices, guided by our separation principle, we propose a novel uncertainty score based on conditional cross-entropy that explicitly quantifies the separation between correct and incorrect predictions in the uncertainty space. The main contributions of this paper are:

- We propose a novel entropy-based UQ score that explicitly quantifies the separation between correct and incorrect predictive distributions using conditional cross-entropy (CCE), resulting in the correctness-aligned uncertainty (CAU) scores account for whether a prediction is correct or not.

- We establish a principled connection between multi-class UQ and binary decision theory, enabling the use of Neyman–Pearson and Bayesian detection principles for structured uncertainty assessment.

- We demonstrate that beyond serving as a metric for evaluating uncertainty performance across systems, the proposed CAU score can also be a tool to guide model design. The dual purposes can be achieved simply by incorporating the CAU score into the objective as a regularizer.

- We conducted extensive experiments to demonstrate how uncertainty separation can be functionally improved with the proposed CAU score, enabling more interpretable and robust model behavior.

## 2 ALIGNING CONFIDENCE WITH CORRECTNESS: A CONDITIONAL CROSS-ENTROPY APPROACH TO UNCERTAINTY QUANTIFICATION

This section introduces a conditional cross-entropy (CCE) approach to UQ that aligns uncertainty estimates with prediction correctness, providing a principled measure for cross-system uncertainty evaluation, improved model calibration and training, and enhanced misclassification detection.

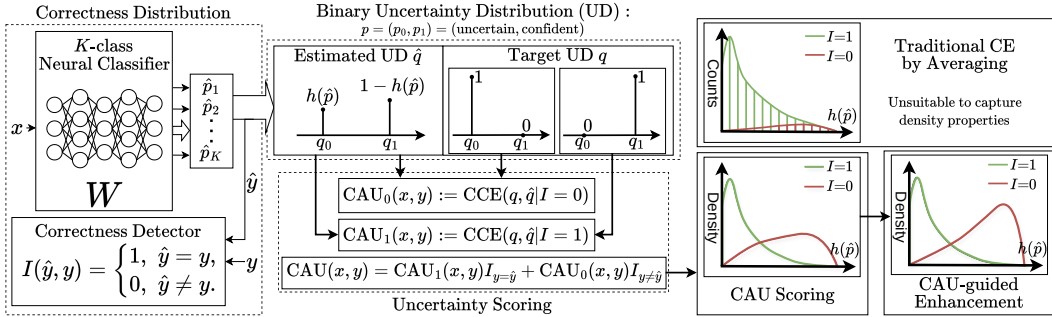

Figure 1: Overview of the correctness-aligned uncertainty (CAU) framework. The pipeline illustrates our approach to uncertainty quantification (UQ) that explicitly aligns model confidence with prediction correctness. Given the output probabilities $\hat{p}$ of a $K$-class neural classifier, we extract the prediction $\hat{y}$ and compute the normalized self entropy (SE) $h(\hat{p})$ to estimate a *binary uncertainty distribution* (UD): $\hat{q} = (1 - h(\hat{p}), h(\hat{p}))$. The correctness indicator $I(\hat{y}, y)$ determines whether $\hat{y}$ matches the ground truth $y$, enabling us to define two *target uncertainty distributions*: $(1, 0)$ for correct predictions (high confidence) and $(0, 1)$ for incorrect predictions (high uncertainty). Conditional cross-entropy (CCE) is used to quantify alignment between the estimated and target uncertainty distributions for correct ($I = 1$) and incorrect ($I = 0$) predictions separately, producing the CAU scores. Population density graphs illustrate that traditional SE metric for UQ fails to enforce distributional separation between correct and incorrect predictions, while our CAU creates clearer separation of the two populations. This framework enables cross-system uncertainty evaluation and uncertainty-aware model improvement via CAU-regularized training, supporting effective calibration, misclassification detection and selective prediction in high-stake applications.

### 2.1 CORRECTNESS-ALIGNED UNCERTAINTY (CAU) SCORING

Consider a standard $K$-class classification problem, where each input $x$ has a ground-truth label $y \in \{1, \ldots, K\}$. A classifier produces a predictive probability distribution $\hat{p} = (\hat{p}_1, \ldots, \hat{p}_K)$, where $\hat{p}_k \geq 0$ and $\sum_{k=1}^{K} \hat{p}_k = 1$. The true label is represented as a one-hot distribution $p = (p_1, \ldots, p_K)$, with $p_y = 1$ for the true class $k = y$ and $p_k = 0$ for $k \neq y$. The cross-entropy (CE) loss measures the dissimilarity between the true and predicted distributions, defined as:

$$\mathrm{CE}(p, \hat{p}) = -\sum_{k=1}^{K} p_k \log \hat{p}_k = \underset{\text{(for one-hot encoded } y)}{-\log \hat{p}_y} . \tag{1}$$

Using one-hot encoding penalizes the model when it assigns low probability to the correct class.

A common metric for quantifying the model's intrinsic uncertainty is the self-entropy (SE) of $\hat{p}$:

$$\mathrm{SE}(\hat{p}) = -\sum_{k=1}^{K} \hat{p}_k \log \hat{p}_k \in [0, \log K], \tag{2}$$

which peaks at $\log K$ for a uniform predictive distribution (maximum uncertainty) and approaches zero when concentrated on a single class (maximum confidence), irrespective of correctness.

While SE reflects the confidence of each decision $\hat{y} = \arg\max_k \hat{p}_k$, it is agnostic to whether that decision is correct. As a result, uncertainty mitigation strategies that simply reduce SE may boost confidence in both correct and incorrect predictions indiscriminately, offering little benefit for detecting classification errors or interpreting model behavior. Further, as a single compact score, particularly in its empirical average form, SE lacks adequate information to reliably compare uncertainty across systems on the same task, as elaborated in Section 2.3 and experimentally verified in Section 4.

These limitations motivate us to revisit UQ for classifiers. Our goal is to design correctness-aligned uncertainty (CAU) scores that account for whether a prediction is correct or not. Such scores not only improve interpretability but also enable accuracy improvement through model refinement and misclassification detection. Figure 1 illustrates the key ideas, which we detail next, while Section 4 demonstrates the usefulness of the introduced UQ scores on algorithmic improvement.

First, to develop interpretable UQ scores, we introduce a *binary uncertainty distribution* $q = (q_0, q_1)$, representing the probabilities of the model being *confident* ($q_0$) or *uncertain* ($q_1$), with $q_0 = 1 - q_1$. Unlike common scalar UQ measures such as SE or variance, which provide only partial summaries and may lack interpretability, the proposed probabilistic representation $q$ captures uncertainty more meaningfully. Our first step is to estimate $q$ for a given input $x$ and classifier output $\hat{p}$. To achieve this, we reinterpret SE values probabilistically by mapping them into $[0, 1]$. Specifically, we normalize $\text{SE}(\hat{p})$ in (2) by its maximum $\log K$, yielding a probability mass value $h(\hat{p})$. Setting $\hat{q}_1 = h(\hat{p})$, the uncertainty distribution can be estimated during training and prediction as

$$\hat{q} = (\hat{q}_0, \hat{q}_1) = (1 - h(\hat{p}), h(\hat{p})), \quad \text{where} \quad h(\hat{p}) = \frac{\text{SE}(\hat{p})}{\log K} \in [0, 1]. \tag{3}$$

Next, to align UQ scores with decision correctness, we recast the model output as a binary decision: correct classification ($I_{y=\hat{y}} = 1$) or incorrect classification ($I_{y=\hat{y}} = 0$), where $I_{(\cdot)}$ is the binary-valued indicator function. The decision follows a *binary correctness distribution* $P = (P_{cor}, P_{incor})$, where $P_{cor} := P(y = \hat{y})$ and $P_{incor} := P(y \neq \hat{y})$. To align confidence with correctness, we define the *binary target uncertainty distribution* by one-hot encoding:

$$q = (q_0, q_1) = \begin{cases} (1, 0), & I_{y=\hat{y}} = 1, \\ (0, 1), & I_{y=\hat{y}} = 0. \end{cases} \tag{4}$$

That is, when the classification is correct ($I = 1$), the model should be confident with $q_0 = 1$; otherwise, it should be maximally uncertain with $q_1 = 1$ for $I = 0$.

Now, with (3) and (4) in place, we define new UQ scores to quantify the dissimilarity between the estimated and target uncertainty distributions using conditional cross-entropy (CCE), resulting in correctness-aligned uncertainty (CAU) quantities as UQ scores per sample $(x, y)$:

$$\begin{aligned} \text{CAU}_1(x, y) &:= \text{CCE}(q, \hat{q} | I = 1) = -\textstyle\sum_{i=0}^{1} q_i \log \hat{q}_i = (-\log \hat{q}_0) I_{y=\hat{y}} = (1 - h(\hat{p})) I_{y=\hat{y}}, \\ \text{CAU}_0(x, y) &:= \text{CCE}(q, \hat{q} | I = 0) = -\textstyle\sum_{i=0}^{1} q_i \log \hat{q}_i = (-\log \hat{q}_1) I_{y\neq\hat{y}} = h(\hat{p}) I_{y\neq\hat{y}}. \end{aligned} \tag{5}$$

They can be expressed in a unifying form as follows:

$$\text{CAU}(x, y) = \text{CAU}_1(x, y) I_{y=\hat{y}} + \text{CAU}_0(x, y) I_{y\neq\hat{y}} = (1 - h(\hat{p})) I_{y=\hat{y}} + \lambda h(\hat{p}) I_{y\neq\hat{y}}. \tag{6}$$

These two CCE-based metrics score uncertainty separately for correct and incorrect decisions, providing richer insights for assessing model reliability and guiding post-decision error detection. This framework departs from conventional UQ using a single averaged score, which fails to separate the distributions of these two decision states. By introducing decision-conditional metrics, our approach connects UQ in multi-class classifiers to binary decision theory, enabling principled uncertainty assessment using established principles such as Neyman-Pearson and Bayesian detection. We elaborate on these decision-theoretic perspectives for structuring effective UQ metrics in Section 2.3.

## 2.2 EMPIRICAL CAU SCORES

Given a dataset, we define the empirical CAU as the statistical expectation of sample-level CAU over the data distribution $\mathcal{D}$. Let $\mathcal{D}_1$ and $\mathcal{D}_0$ denote the subsets of correctly and incorrectly classified samples, with $\mathcal{D} = \mathcal{D}_1 \cup \mathcal{D}_0$, and sizes $N_1 = |\mathcal{D}_1|$ and $N_0 = |\mathcal{D}_0|$ respectively. We define

$$\text{CAU}_1 := \mathbb{E}_{(x,y)\sim\mathcal{D}_1} \text{CAU}_0 := \mathbb{E}_{(x,y)\sim\mathcal{D}_0}[\text{CAU}_0(x, y)], \tag{7}$$

In practice, with $N$ labeled samples $\{(x_i, y_i)\}_{i=1}^N$, an expectation is approximated by the sample average. The two CAU metrics in (7) are approximated as

$$\begin{aligned} \ell_1 := \widehat{\text{CCU}}_1 &= \frac{1}{N_1} \sum_{(x_i,y_i)\in\mathcal{D}_1} \text{CCU}_1(x_i, y_i) = \frac{1}{N_1} \sum_{(x_i,y_i)\in\mathcal{D}_1} 1 - h(\hat{p}(x_i)), \\ \ell_0 := \widehat{\text{CCU}}_0 &= \frac{1}{N_0} \sum_{(x_i,y_i)\in\mathcal{D}_0} \text{CCU}_0(x_i, y_i) = \frac{1}{N_0} \sum_{(x_i,y_i)\in\mathcal{D}_0} h(\hat{p}(x_i)). \end{aligned} \tag{8}$$

## 2.3 DECISION-THEORETIC FORMULATIONS FOR CAU

Both CAU metrics $\ell_1$ and $\ell_0$ in (8) take the form of negative log-likelihoods and can be interpreted as test statistics of a binary hypothesis test: $H_1$: $I_{y=\hat{y}}$ (correct decisions) vs. $H_0$: $I_{y=\hat{y}}$ (incorrect decisions). This formulation introduces a novel decision-theoretic perspective on UQ, allowing us to leverage established principles of binary decision theory for principled assessment in multi-class classification. Our goal is to design UQ scores that not only quantify uncertainty but also serve as effective training costs or misclassification detectors, thereby improving predictive accuracy.

From a Bayesian risk perspective, a meaningful UQ cost is the overall expected cost, which accounts for the prior probabilities of the classification correctness hypotheses $H_i$ and the associated uncertainty costs $C_i$, $i = 1, 0$. Given conditional CE, the Bayesian cost of uncertainty is defined as

$$\mathrm{CAU}(\ell_1, \ell_0; C_1, C_0) = C_1 \, \mathrm{CAU}_1 \, P(H_1) + C_0 \, \mathrm{CAU}_0 \, P(H_0) \overset{\mathcal{D}}{\approx} C_1 \, \ell_1 \, \frac{N_1}{N} + C_0 \, \ell_0 \, \frac{N_0}{N}. \quad (9)$$

When $C_0 = C_1 = 1$, (9) reduces to empirical average of $\mathrm{CAU}(x, y)$ in (6):

$$\mathrm{CAU}(\ell_1, \ell_0; 1, 1) = \frac{1}{N} \sum_{(x_i, y_i) \in \mathcal{D}} \mathrm{CCU}(x, y) = \frac{1}{N} \sum_{(x_i, y_i) \in \mathcal{D}_1} \mathrm{CCU}_1(x_i, y_i) + \frac{1}{N} \sum_{(x_i, y_i) \in \mathcal{D}_0} \mathrm{CCU}_0(x_i, y_i)$$

$$= \ell_1 \frac{N_1}{N} + \ell_0 \frac{N_0}{N}, \quad (10)$$

which resembles the common practice of averaging uncertainty scores across samples. However, this formulation is fundamentally ill-suited for neural classifiers. In high-accuracy models, correct predictions vastly outnumber incorrect ones ($N_1 \gg N_0$), so the prior $P(H_1)$ overwhelms the contribution of $P(H_0)$. This severe imbalance in the binary correctness distribution $I$ is inherent to accurate classifiers, even when the underlying dataset is balanced across the $K$ classes. Consequently, the resulting score in (10) is driven largely by model accuracy, undermining its value as an uncertainty-centric metric. This equivalence to the standard averaging approach further explains why conventional UQ methods fail to provide meaningful insight for cross-system comparison.

To enable meaningful cross-system uncertainty evaluation, we counteract the inherent imbalance between correct and incorrect decision samples by reducing the influence of model-specific accuracy priors, so that CAU focuses on uncertainty rather than accuracy. A simple yet effective strategy is to set $C_1 = N/N_1$ and $C_0 = \lambda N/N_0$, which transforms the Bayesian risk in (9) to a $\lambda$-scaled CAU:

$$\mathrm{CAU}_\lambda(\ell_1, \ell_0) := \mathrm{CAU}(\ell_1, \ell_0; C_1 = \frac{N}{N_1}, C_0 = \lambda \frac{N}{N_0}) = \ell_1 + \lambda \ell_0. \quad (11)$$

Interestingly, (11) not only neutralizes the inherent non-iid imbalance between correct and incorrect decisions in accurate classifiers, but also reveals an natural connection to the Neyman–Pearson (NP) framework in classical (non-Bayesian) binary decision theory. Since the CCE metrics correspond to negative log-likelihoods under the two hypotheses, the NP principle formulates the test as

$$\min \ell_1, \qquad \text{s.t.} \quad \ell_0 \le \Gamma_0 \quad (12)$$

where $\Gamma_0$ specifies allowable CCE loss under $H_0$. By introducing a Lagrangian multiplier $\lambda$, this constrained optimization in (12) recovers the same unconstrained cost structure (11). Thus, our formulation can be interpreted both as Bayesian risk minimization with tailored costs and as an NP test that avoids reliance on priors, an attractive property given the severe imbalance inherent in classifier correctness outcomes. The scaled CAU in (11) directly balances the tradeoff between incorrect and correct decisions, which is an appealing property for principled uncertainty quantification.

## 3 UNCERTAINTY-AWARE LEARNING WITH CAU REGULARIZED LOSS

Beyond serving as an evaluation metric, the proposed CAU score can also effectively guide model design, leveraging its cross-system evaluation capability that is lacking in conventional UQ scores. Model training often prioritizes accuracy while neglecting uncertainty, a bias that drives models toward overconfident predictions rather than faithful uncertainty expression, exemplified by phenomena such as LLM hallucinations (Kalai et al. (2025)). To address this, we develop uncertainty-aware learning by embedding CAU into the training objective, aligning uncertainty estimates with prediction correctness. This alignment improves both calibration and predictive accuracy.

Formally, we define a neural classifier $\hat{y}_\theta$, parametrized by $\theta$, trained with a standard cross-entropy loss for softmax classification augmented by a CAU regularization term:

$$\mathcal{L} = \text{CE}(\hat{y}, y) + \alpha \, \text{CAU}_\lambda(\hat{y}, y). \tag{13}$$

Optimizing this objective explicitly encourages the model to reduce uncertainty for correct predictions and increase it for incorrect ones, creating a clearer separation between the two. This separation directly benefits post-classification tasks such as selective prediction and misclassification detection during inference, even when the correctness of predictions is unknown. Large values of the uncertainty probability estimate $h(\hat{p})$ in (3)) indicate likely misclassification. In this way, CAU serves dual purposes: a principled metric for (cross-system) uncertainty quantification and misclassification detection, and a practical training regularizer for training more reliable neural classifiers.

## 4 EXPERIMENTS

### 4.1 EXPERIMENTAL DESIGN

In accordance with our two main contributions, this section is organized into two parts. The first part evaluates the effectiveness of the proposed CAU score, highlighting its advantages over existing misclassification detection metrics. The second part illustrates that the CAU augmented loss function enhances model uncertainty capabilities by reducing uncertainty for correct predictions and increase it for incorrect ones.

#### 4.1.1 EFFECTIVENESS VALIDATION SETUP

The effectiveness of CAU lies in its ability to provide reliable uncertainty assessment for models with varying accuracy levels and to clearly distinguish between higher and lower degrees of uncertainty. To examine this property, we design experiments on the EMNIST and MNIST datasets. On EMNIST (62 classes, relatively challenging) (Cohen et al., 2017), we adopt a simple multilayer perceptron (MLP, 784-128-62), which yields an overall accuracy close to 50%. On MNIST (LeCun et al., 2010), we use the classical LeNet-5 architecture (Lecun et al., 1998), which achieves an accuracy close to 99%. By setting up two distinct accuracy regimes, one with relatively balanced proportions of correct and incorrect samples and the other with highly imbalanced proportions, we are able to test whether CAU remains valid under different degrees of sample balance.

To further generate models with varying levels of uncertainty performance, we employ dropout. Prior studies have shown that better calibration, namely a closer alignment between predicted probabilities and true class frequencies, leads to more reliable uncertainty estimation (He et al., 2025), and dropout has been widely recognized as a technique that improves calibration (Srivastava et al., 2014; Carratino et al., 2022; Mukhoti et al., 2020). As the dropout rate increases within a certain range where dropout reduces overfitting without harming convergence, the uncertainty quality of the model is expected to improve. We therefore vary the dropout rate to obtain models spanning different uncertainty levels.

For comparison, we consider three post-processing metrics that are widely used in misclassification detection research: AUROC, AUPR-Error, and AUPR-Correct. AUROC measures the overall ranking capability of a model in separating correct from incorrect predictions by integrating the receiver operating characteristic curve. AUPR-Error, which treats incorrect predictions as the positive class, directly reflects the ability to discover rare errors. AUPR-Correct, by contrast, treats correct predictions as the positive class and evaluates the reliability of confirming normal cases.

#### 4.1.2 TRAINING OBJECTIVE ENHANCEMENT SETUP

To investigate the effect of incorporating CAU into the training objective on uncertainty separation between correct and incorrect samples and on the performance of post-processing misclassification detection, we design the following experiment. We consider the classification task on the moderately challenging Kuzushiji-MNIST dataset (Clanuwat et al., 2018) using a simple multilayer perceptron (MLP, 784-128-10), to get an accuracy of about 70%-85%. The baseline model is trained with the standard cross-entropy loss, while the experimental model is trained with an augmented loss that adds the CAU term.

We then evaluate the two models on post-processing misclassification detection tasks. Specifically, we compare AUROC, AUPR-Error, and AUPR-Correct, together with correct–error distribution plots and CAU values, to examine whether the augmented loss leads to clearer separation of uncertainty and improved detection performance.

## 4.2 RESULTS AND ANALYSIS

### 4.2.1 EFFECTIVENESS IN CROSS-SYSTEM COMPARISON

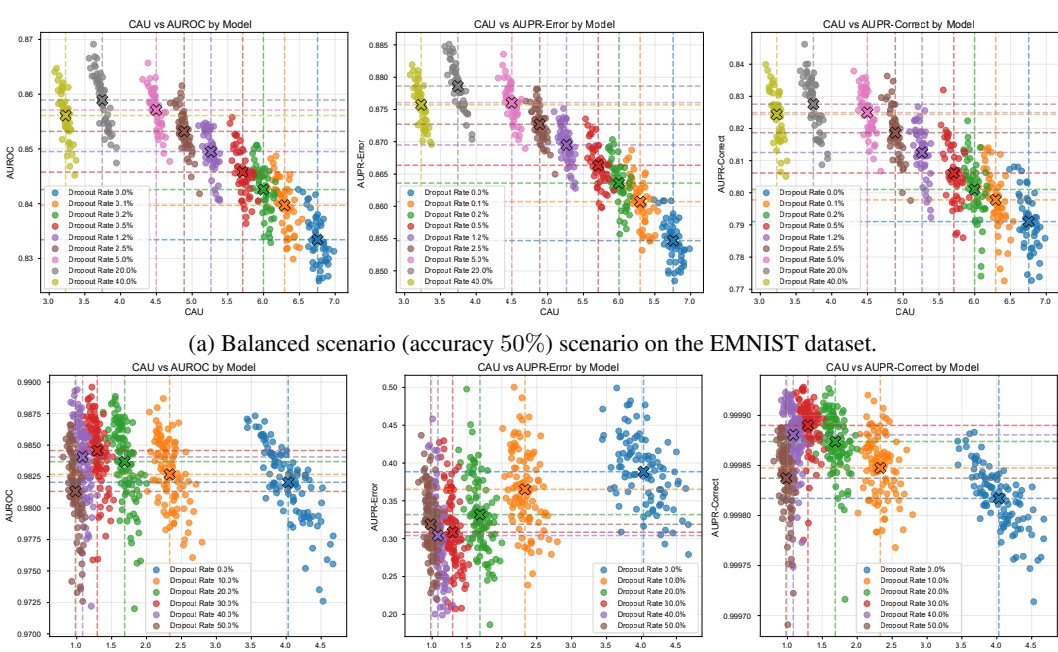

(a) Balanced scenario (accuracy $50\%$) scenario on the EMNIST dataset.

(b) Imbalanced scenario (accuracy $99\%$) on the MNIST dataset.

Figure 2: CAU Score vs. traditional metrics comparison. Each point represents test set results from models trained under fixed architecture-dataset combinations with different random seeds (50 seeds per dropout rate for balanced scenarios, 100 seeds per dropout rate for high accuracy scenarios). Each color represents a different dropout rate. (a) Balanced scenarios show clear negative correlations demonstrating metric effectiveness. (b) High accuracy scenarios show traditional metrics with inconsistent relationships while CAU Score maintains clear trends.

Table 1: Correlation Analysis: Dropout Rate vs. Uncertainty Metrics

| Metric | Balanced (EMNIST) | | | Imbalanced (MNIST) | | |
|---|---|---|---|---|---|---|
| | $r$ | $R^2$ | $p$ | $r$ | $R^2$ | $p$ |
| CAU | -0.839 | 0.704 | $< 0.001^{***}$ | -0.889 | 0.790 | $< 0.001^{***}$ |
| AUROC | 0.523 | 0.273 | $< 0.001^{***}$ | 0.022 | 0.000 | 0.595 |
| AUPR-Error | 0.553 | 0.306 | $< 0.001^{***}$ | -0.461 | 0.212 | $< 0.001^{***}$ |
| AUPR-Correct | 0.519 | 0.269 | $< 0.001^{***}$ | 0.259 | 0.067 | $< 0.001^{***}$ |

**Balanced Scenario Analysis:** Figure 2a presents the results under the balanced scenario, where model accuracy is close to 50%. For different dropout rates, the outputs include the CAU score together with post-processing metrics AUROC, AUPR-Error, and AUPR-Correct. As the dropout rate increases, the CAU score decreases, while the post-processing metrics improve correspondingly, showing a clear negative correlation between CAU and these metrics. This observation is consistent with the Bayesian risk analysis, confirming that CAU effectively captures variations in uncertainty quality under balanced sample conditions. The regression analysis between various metrics and Dropout Rate (DR), shown in Table 1, demonstrates the significant superiority of UQ Score. With $R^2 = 0.704$, it ranks first among all comparison metrics with a clear advantage. Specifically, UQ

Score's $R^2$ value is 130% higher than the second-best metric AUPR-Error ($R^2 = 0.306$), showing stronger association and predictability. Meanwhile, UQ Score's $R^2$ value is also more than twice that of AUROC ($R^2 = 0.273$) and AUPR-Correct ($R^2 = 0.269$).

**Imbalanced Scenario Analysis:** Figure 2b shows results under the imbalanced scenario, where model accuracy is close to 99%. Across different dropout rates, the CAU score exhibits a clear decreasing trend, reflecting improved uncertainty quality as dropout increases. In contrast, the post-processing metrics AUROC, AUPR-Error, and AUPR-Correct fail to capture these variations. Their values are dominated by the overwhelming proportion of correct samples, which masks the underlying uncertainty differences. As a result, in imbalanced settings these traditional metrics lose their ability to quantify uncertainty performance, whereas CAU remains sensitive and reliable.

**Robustness Analysis:** An additional observation is that traditional metrics are highly sensitive to random seed variations, resulting in substantial overlap in the AUROC dimension, as illustrated in Figure 2. By contrast, CAU is much less affected by this issue. The correlation analysis in Table 1 further confirms its robustness: with $R^2 = 0.790$, changes in dropout rate explain 79% of the variance in CAU, accompanied by a highly significant ($p < 0.001$) and strong negative correlation. This makes CAU the most reliable measure for evaluating the effect of dropout across systems. In comparison, AUPR-Error also shows a statistically significant correlation ($p < 0.001$) but with a much weaker $R^2 = 0.212$, indicating limited stability and predictive power. AUPR-Correct has an even weaker relationship ($R^2 = 0.067$), and AUROC exhibits no statistically significant correlation with dropout rate ($p = 0.595$). Taken together, these results demonstrate that CAU Score provides a substantially more reliable and sensitive assessment of dropout effects than misclassification metrics.

### 4.2.2 TRAINING OBJECTIVE ENHANCEMENT

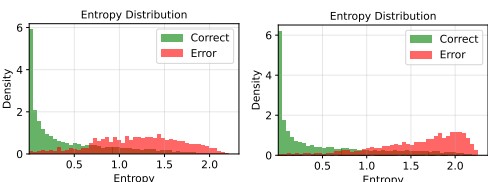

Figure 3: Entropy distributions: baseline (left) and CAU Score augmented loss model (right). Modified model shows clearer separation between correct and incorrect predictions.

Table 2: Performance Comparison: Baseline vs. CAU Score augmented Loss Function on Kuzushiji-MNIST (50 seeds each).

| Metric | Baseline | Modified |
|---|---|---|
| Accuracy | **0.8008 ± 0.0050** | 0.7493 ± 0.0078 |
| AUROC | 0.8732 ± 0.0032 | **0.8977 ± 0.0039** |
| AUPR-Correct | **0.9657 ± 0.0015** | 0.9646 ± 0.0022 |
| AUPR-Error | 0.5881 ± 0.0119 | **0.7144 ± 0.0147** |
| Correct Entropy | 0.4341 ± 0.0211 | **0.5386 ± 0.0182** |
| Error Entropy | 1.2564 ± 0.0510 | **1.6410 ± 0.0511** |
| CAU | **0.9908 ± 0.0198** | 0.7988 ± 0.0148 |

Figure 3 compares the uncertainty distributions of correct and incorrect samples under the standard cross-entropy loss and the CAU-augmented loss. The results clearly show that the CAU-augmented model produces a more pronounced separation between the two distributions. This wider separation facilitates the choice of uncertainty thresholds that more effectively distinguish between correct and incorrect predictions.

Table 2 further compares post-processing misclassification detection performance between the two training schemes. The CAU-augmented model incurs a modest decrease of about 5.15% in classification accuracy compared to the baseline. However, the modified model increases entropy separation significantly. In the baseline model, correct predictions have an average entropy of 0.43 while incorrect predictions have an average entropy of 1.26, yielding a separation gap of 0.82. For the modified model, correct predictions maintain a lower average entropy of 0.54 while incorrect predictions show a much higher value (1.64), expanding the gap to 1.10. This wider separation makes it easier to set uncertainty thresholds to distinguish between correct and incorrect predictions. Thus, it achieves a substantial improvement in error detection metrics, most notably AUPR-Error, which increases by 21.5% (from 0.5881 to 0.7144). This demonstrates that the CAU-augmented loss substantially enhances the model's ability to identify its own errors, representing a necessary and worthwhile trade-off in high-risk scenarios.

To demonstrate the practical application potential of our modifications, we evaluate the operation performance under high-reliability deployment constraints. We assume a high-risk scenario (such as large-scale early cancer screening) where effective screening programs must have extremely high sensitivity, typically requiring over 99% Negative Predictive Value (Trevethan, 2017). To simulate

this high-level safety requirement, we set the system's reliability target to 99.95%:

$$\text{Reliability} = \frac{N_{\text{total}} - N_{\text{missed errors}}}{N_{\text{total}}} \geq 0.9995. \tag{14}$$

This constraint ensures that the proportion of silent failure cases does not exceed 0.05%. Under this strict risk constraint, we evaluate the operational costs (i.e. human review rate) both models must pay to achieve this goal, with results presented in Table 3.

Table 3: Performance Under 99.95% Reliability Constraint

| Metric | Baseline | Modified |
|---|---|---|
| Original Accuracy (%) | $\mathbf{80.08 \pm 0.50}$ | $74.93 \pm 0.79$ |
| Total Review Rate (%) | $86.91 \pm 2.17$ | $\mathbf{81.06 \pm 1.77}$ |
| Automatic Pass Rate (%) | $16.28 \pm 2.70$ | $\mathbf{25.20 \pm 2.28}$ |
| Invalid Review Cost | $6703.8 \pm 218.2$ | $\mathbf{5604.2 \pm 161.8}$ |

The modified model demonstrates decisive advantages. Its total review rate is 5.85% points lower than the baseline model, meaning it can reduce nearly 600 review cases for doctors or reviewers when screening a cohort of 10,000 people. More importantly, the modified model's automatic pass rate reaches 25.2%, compared to the baseline model's 16.3%, representing an improvement of over 54%. Notably, even accounting for the trade-off in raw accuracy (5.15%), the enhanced automation capabilities lead to net reductions in human workload when human oversight is incorporated. This indicates that the modified model can identify and automatically process normal samples with higher confidence.

## 5 CONCLUSION AND FUTURE WORK

In this work, we addressed fundamental challenges in uncertainty quantification for neural classifiers, particularly the limitations of conventional UQ metrics under imbalanced accuracy regimes. We introduced the correctness-aligned uncertainty (CAU) score, which explicitly separates uncertainty for correct and incorrect predictions, thereby providing a more faithful reflection of model reliability. Theoretical analysis established its grounding in decision theory, while experimental studies verified the effectiveness of CAU and demonstrated its clear advantages over existing postprocessing metrics across models with varying accuracy levels and training objectives.

Future research may extend CAU to more complex architectures and large-scale applications, e.g., federated learning where cross-system comparability is critical. Another promising avenue is the application of CAU in high-stakes domains such as healthcare, transportation, and finance, where uncertainty-aware training can translate into tangible operational benefits.

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

## A ADDITIONAL VALIDATION OF CAU EFFECTIVENESS

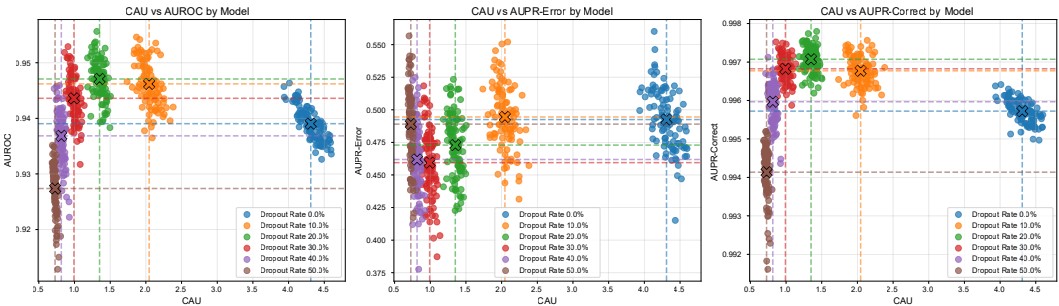

Figure 4: High accuracy $(91\% - 96\%)$ on the Kuzushiji-MNIST dataset.

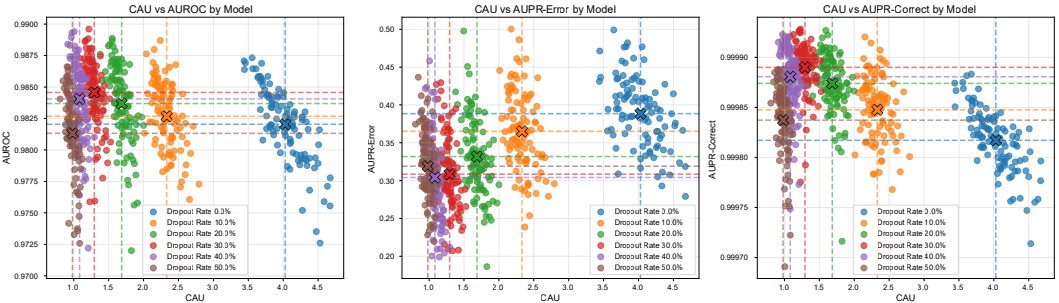

Figure 5: High accuracy (approaching $85\% - 91\%$) on the Fashion-MNIST dataset.

To further validate the generalizability of our findings beyond the MNIST dataset, we conducted extensive experiments on Fashion-MNIST (Xiao et al., 2017) and Kuzushiji-MNIST datasets using the same LeNet-5 architecture. These additional experiments demonstrate that the limitations of traditional uncertainty quantification metrics under high-accuracy conditions are not dataset-specific phenomena but represent fundamental issues affecting cross-system uncertainty evaluation.

### A.1 EXTENDED DATASET ANALYSIS

**Fashion-MNIST Results:** Using LeNet-5 on Fashion-MNIST, we achieved accuracy levels ranging from 85% to 91%. The correlation analysis between dropout rate and various uncertainty metrics

reveals consistent patterns with our MNIST findings. As shown in Figure 5, CAU maintains the strongest relationship with dropout rate ($R^2 = 0.714$, $r = -0.845$, $p < 0.001$), significantly outperforming traditional metrics. AUROC shows moderate correlation ($R^2 = 0.421$), while AUPR-Error exhibits weak correlation ($R^2 = 0.154$), and AUPR-Correct demonstrates intermediate performance ($R^2 = 0.362$).

**Kuzushiji-MNIST Results:** On Kuzushiji-MNIST, with accuracy ranging from 91% to 96%, the superiority of CAU becomes even more pronounced. The correlation analysis shows CAU achieving the highest explanatory power ($R^2 = 0.738$, $r = -0.859$, $p < 0.001$). Traditional metrics show substantially degraded performance: AUROC drops to $R^2 = 0.314$, AUPR-Error collapses to $R^2 = 0.047$, and AUPR-Correct achieves only $R^2 = 0.225$. These results, presented in Figure 4, demonstrate that as model accuracy increases, traditional metrics become increasingly unreliable for uncertainty assessment.

## A.2 CROSS-DATASET CONSISTENCY ANALYSIS

Table 4 summarizes the correlation analysis across all three datasets. The consistent superiority of CAU across different data distributions and accuracy ranges validates our theoretical analysis about the fundamental limitations of traditional UQ metrics in imbalanced scenarios.

Table 4: Cross-Dataset Correlation Analysis: Dropout Rate vs. Uncertainty Metrics

| Metric | MNIST (99% acc) | | | Fashion-MNIST (85-91% acc) | | | Kuzushiji-MNIST (91-96% acc) | | |
|---|---|---|---|---|---|---|---|---|---|
| | $r$ | $R^2$ | $p$ | $r$ | $R^2$ | $p$ | $r$ | $R^2$ | $p$ |
| CAU | -0.889 | 0.790 | $< 0.001^{***}$ | -0.845 | 0.714 | $< 0.001^{***}$ | -0.859 | 0.738 | $< 0.001^{***}$ |
| AUROC | 0.022 | 0.000 | 0.595 | -0.649 | 0.421 | $< 0.001^{***}$ | -0.560 | 0.314 | $< 0.001^{***}$ |
| AUPR-Error | -0.461 | 0.212 | $< 0.001^{***}$ | -0.392 | 0.154 | $< 0.001^{***}$ | -0.216 | 0.047 | $< 0.001^{***}$ |
| AUPR-Correct | 0.259 | 0.067 | $< 0.001^{***}$ | -0.602 | 0.362 | $< 0.001^{***}$ | -0.474 | 0.225 | $< 0.001^{***}$ |

# B ADDITIONAL VALIDATION OF THE ROBUSTNESS OF CAU-AUGMENTED LOSS EFFECTIVENESS

To demonstrate the consistency and robustness of our CAU-augmented training approach, we conducted experiments across multiple random seeds and present detailed entropy distribution comparisons. Figure 6 shows entropy distribution comparisons for five different random seeds, where each comparison displays the baseline model (left) and the CAU-augmented model (right) trained with identical initialization seeds.

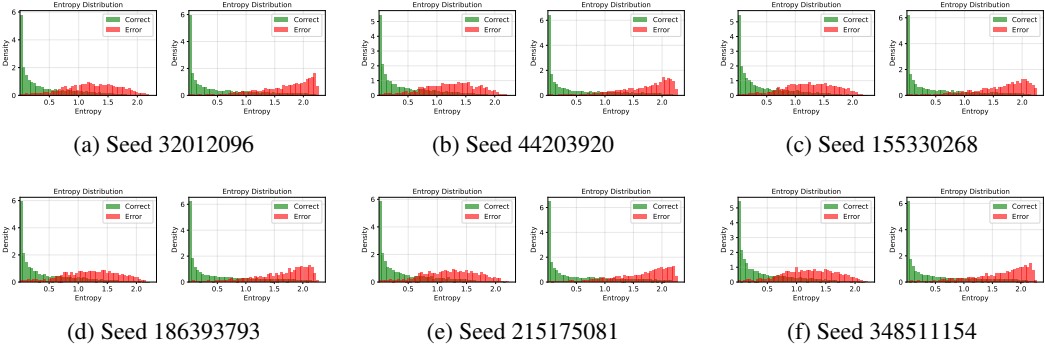

(a) Seed 32012096      (b) Seed 44203920      (c) Seed 155330268

(d) Seed 186393793      (e) Seed 215175081      (f) Seed 348511154

Figure 6: Entropy distribution comparisons across multiple random seeds. Each pair shows baseline model (left) and CAU-augmented model (right) trained with identical seeds. The consistent pattern across all seeds demonstrates that CAU-augmented training reliably produces clearer separation between correct and incorrect predictions regardless of random initialization.

The consistent improvement across all tested seeds validates that the enhanced separation between correct and incorrect entropy distributions emerges consistently across different random initializa-

tions, suggesting that CAU-augmented training exhibits stable convergence properties and reliable uncertainty enhancement across different optimization trajectories.

## C  ETHICS STATEMENT

This work adheres to the ICLR Code of Ethics. In this study, no human subjects or animal experimentation was involved. All datasets used, were sourced in compliance with relevant usage guidelines, ensuring no violation of privacy. We have taken care to avoid any biases or discriminatory outcomes in our research process. No personally identifiable information was used, and no experiments were conducted that could raise privacy or security concerns. We are committed to maintaining transparency and integrity throughout the research process.

## D  LLM USAGE

Large Language Models (LLMs) were used to aid in the writing and polishing of the manuscript. Specifically, we used an LLM to assist in refining the language, improving readability, and ensuring clarity in various sections of the paper. The model helped with tasks such as sentence rephrasing, grammar checking, and enhancing the overall flow of the text.

It is important to note that the LLM was not involved in the ideation, research methodology, or experimental design. All research concepts, ideas, and analyses were developed and conducted by the authors. The contributions of the LLM were solely focused on improving the linguistic quality of the paper, with no involvement in the scientific content or data analysis.

The authors take full responsibility for the content of the manuscript, including any text generated or polished by the LLM. We have ensured that the LLM-generated text adheres to ethical guidelines and does not contribute to plagiarism or scientific misconduct.

