# OpenReview forum: "Uncertainty Quantification Revisited: Conditional Cross-Entropy for Neural Classifiers"
_ICLR.cc/2026/Conference — Submitted to ICLR 2026_

### Official Review · Reviewer_4ZrA · 2025-10-23

**Soundness:** 2
**Presentation:** 2
**Contribution:** 2
**Rating:** 2
**Confidence:** 4

**Summary:**

The paper introduces CAU scores, a measure of (misclassification) uncertainty quantification that allows for an explicit account of whether predictions are correct or incorrect. Authors also demonstrate that incorporating CAU into the training objective enhances the separation of uncertainty between correct and incorrect predictions and improves post-processing misclassification detection.

**Strengths:**

1) The paper is easy to follow

**Weaknesses:**

1) Writing.

Some claims in the paper refer to "theoretical guarantees", but it is never explicitly stated what these guarantees are, and whether CAU provides any of them. Currently, the introductory section is presented in a way that the CAU will be a theoretically grounded method, and these theoretical guarantees are sort of a "contrast" to existing approaches. But what these guarantees are is never mentioned or discussed.


For example,
> In line 040, "...widely studied thanks to their theoretical guarantees"

> In lines 055-056: "However, these methods either fail to produce direct uncertainty estimates or lack strong theoretical guarantees."

> In lines 057-058: "self-entropy (SE) based UQ methods that directly compute the Shannon entropy of the softmax output enjoy both solid theoretical foundation and computational efficiency".

What are the guarantees provided by entropy, applied to softmax output? Why doesn't entropy, applied to the second-order distribution predicted by an Evidential deep learning model (mentioned in the paper earlier, line 054), provide the same "degree" of guarantees?


2) Particular design choice of the method

Authors criticize other approaches for the lack of theoretical guarantees. However, the methods proposed in this paper are a pure heuristic. It starts with assigning a probability of incorrect output. The authors propose setting the probability of incorrect prediction to the normalized value of the predicted entropy. Why is it so? Why not put it to the 1-max prob or divide the variance of the most confident label by the maximal variance value? Multiple design choices could be made, and the authors do not justify the one they use.

3) Typo in derivations:

In Equation 5, which is likely the central equation for the method, the authors consider the cross-entropy between the one-hot vector q (an unknown indicator) and the newly formed vector of probabilities of correctness, whose elements are introduced in Equation 3. However, the logarithm is missing in the derivation.

This issue of a missing logarithm appears in equation 6. Moreover, in equation 6, there are extra indicators (although I understand that I^2 = I, but it still seems redundant).

In equation 13, some parts of the CE will be taken into account twice. Will it still be a proper scoring rule that has a theoretical minimum in the correct point (so that the global minimizer should coincide with the theoretical distribution p(y | x) ?)


4) Positioning.

The title of the paper promises a new measure for uncertainty quantification. However, this CAU measure applies only to in-distribution related uncertainty measures; therefore, it ignores out-of-distribution related downstream problems. This should be explicitly mentioned.

**Questions:**

1) What is meant by line 046 "In classification, however, the 'variance' term loses its intuitive meaning"?

2) Is equation 13 a proper scoring rule? Does the global minimizer of this loss function coincide with $p(y
mid x)$?

---

### Official Review · Reviewer_uELJ · 2025-10-24

**Soundness:** 2
**Presentation:** 2
**Contribution:** 1
**Rating:** 2
**Confidence:** 4

**Summary:**

In the paper, a formula is proposed to combine binary classification accuracy and model uncertainty. In my understanding, the formula is normalized self-entropy in case of correct prediction and 1-the same otherwise. While it is possible that I overlook something, the notations seem unnecessarily complex. Also, the paper mentions multi-class and I was expecting this as the reason for using value pairs q, 1-q for the binary case,  I could not find how multi-class is handled.

The limitation of the experiments is that there are no baselines from other research, and too few data sets are used.

While UQ is an important problem, I find the technical contribution both theoretically and experimentally insufficient for ICLR.

**Strengths:**

+ UQ is an important problem
+ Combination of uncertainty and correctness of the prediction in one formula
+ Experiments show consistency of the proposed metric

**Weaknesses:**

- In my understanding, the formula is fairly simple, and the way it is introduced is unnecesarily notation heavy
- Too few data sets are used in the experiments
- No comparison with other baseline UQ methods

**Questions:**

Am I right that CAU is SE/logk if the prediction is correct and 1-SE/logk otherwise?  What is the use of carrying (q,1-q) as pairs? Couldn't the notation be simplified?

My impression is that the discussion around equations (11-12) is disconnected from the definition of SE in CAU and only consider class imbalance mitigation for any function. Or are the underlying logits used more specifically?

How would the proposed method compare to other, otherwise also simple, methods such as True class probability [Corbière, C., Thome, N., Bar-Hen, A., Cord, M., & Pérez, P. Addressing Failure Prediction by Learning Model Confidence. NeurIPS 2019]

Multi-class is mentioned in the paper. Which formula can handle multiclass classification?

Can you handle the case when there are multiple correct answers, see [Abbasi Yadkori, et al. "To believe or not to believe your llm: Iterative prompting for estimating epistemic uncertainty." NeurIPS 2024]?

---

### Official Review · Reviewer_Rygi · 2025-10-31

**Soundness:** 2
**Presentation:** 2
**Contribution:** 2
**Rating:** 2
**Confidence:** 4

**Summary:**

The paper focuses on quality measures for uncertainty quantification (UQ) methods. Specifically, the authors propose a new entropy-based UQ metric called the Correctness-Aligned Uncertainty (CAU) measure, which explicitly accounts for whether predictions are correct or incorrect using conditional cross-entropy. This measure can be used not only as a final evaluation score but also incorporated into the training procedure as an additional loss regularization term, further enhancing the applicability of the CAU score. The paper also presents empirical experiments demonstrating the effectiveness of the CAU score across several models, as well as the performance improvements achieved when models are trained with this regularization.

**Strengths:**

1. Well-structured presentation with a strong and clearly articulated motivation for investigating such a universal uncertainty measure.
2. Comprehensive description of the proposed approach, including its theoretical connections to the Neyman–Pearson and Bayesian detection principles for structured uncertainty assessment.

**Weaknesses:**

1. The core idea of the paper is not entirely novel. Several prior works have already proposed measures that can be used as loss regularization terms to improve uncertainty estimation [1, 2]. Moreover, [1] introduced a new and now well-established misclassification detection metric, distinct from ROC-AUC, which has been successfully applied in several subsequent studies [3].
2. The experimental design is quite limited. The authors consider only a single domain and evaluate two small models on two standard, yet almost outdated, tasks. In contrast, recent works have explored a much broader range of domains and utilized modern architectures for evaluation [4].
3. The description of the proposed method is unclear. Although MC Dropout is a well-established uncertainty quantification technique, its specific usage within the current experimental setup is not adequately explained. For instance, Equation (2) and the subsequent analysis are based on a single predictive distribution and do not appear to consider multiple stochastic forward passes.
4. Figure 2 presents results for a dropout rate of zero. In such a case, it is unclear where the diversity originates. Does it represent identical scores across samples for a single random seed?
5. The main outcomes of the paper are not clearly articulated. The authors claim that CAU is highly correlated with the dropout rate, but it remains unclear why this correlation would imply that CAU is a superior measure.
6. Table 2 shows that the modified model suffers a significant decrease in predictive performance, which could be critical for downstream applications. Although Table 3 suggests that the modified model performs better under high-risk scenarios, the paper does not clearly define how these high-risk scenarios are evaluated.
7. Equation (11) indicates that CAU depends on the hyperparameter $\lambda$. However, the value or tuning process for $\lambda$ is not reported in the experimental section.

[1] The Art of Abstention: Selective Prediction and Error Regularization for Natural Language Processing (Xin et al., ACL-IJCNLP 2021) \
[2] Mitigating Uncertainty in Document Classification (Zhang et al., NAACL 2019) \
[3] Uncertainty Estimation of Transformer Predictions for Misclassification Detection (Vazhentsev et al., ACL 2022) \
[4] Nonparametric Uncertainty Quantification for Single Deterministic Neural Network. (Kotelevskii et al. NeurIPS 2022)

**Questions:**

1. Why does a high correlation with the dropout rate indicate that the proposed measure is superior?
2. How was the high-risk scenario evaluated?

---

### Official Review · Reviewer_FUEm · 2025-11-01

**Soundness:** 1
**Presentation:** 2
**Contribution:** 2
**Rating:** 2
**Confidence:** 4

**Summary:**

A new entropy based score for quantifying the uncertainty of a model is proposed, correctness aligned uncertainty (CAU) that encourages incorrect predictions to have higher entropy and correct ones to have lower entropy, thereby separating the behaviour for incorrect and correct examples. The connection to binary decision theory is highlighted. CAU is also appended to the loss function as a regularization term, and the subsequent models seem to show better class-imbalance aware performance in some datasets. CAU correlates better with dropout percentages within models compared to other misclassification detection measures, and the authors use this to highlight its superiority w.r.t classical measures.

**Strengths:**

- The idea of "increasing" the entropy of the predictions when it is incorrect is interesting.
- Explanation of the idea and overall writing is good.

**Weaknesses:**

The idea of using entropy of predicted softmax output is indeed a very old one, but usually it has always been applied in a manner such as to reduce the uncertainty of the outputs of the network during training (e.g. Shamsi 2023). This leads to consistent behaviour where the loss terms aren't necessarily directly at odds with each other. However, here the authors decrease entropy only in the correct case and do the reverse in the incorrect case. For me fundamentally, there is a very serious instability issue that may arise, as the model's correct/incorrect is often dependent on how far into the optimization the model itself is. Here, such an objective risks premature assessment of incorrectness thereby forcing the model to be uncertain on examples where it could easily have fit given a few more epochs. It may make more sense to do so after having trained the model for a significant number of steps until it reaches near-convergence and then impose this, but then there's also the risk of "unlearning" that comes at that stage, which will be perhaps dependent on the $\lambda$ term that controls how much we impose the "entropy maximization" philosophy. Either way, the 1-entropy term is clearly fundamentally at odds with the overall optimization, and this may lead to some instabilities.

Although the paper discusses "uncertainty quantification" (UQ), it primarily only tests misclassification detection tasks. Typically a better uncertainty quantification measure must align itself with the true uncertainty of the model across many predictions. This leads to the problem of calibration error based approaches, which look into whether the predicted model uncertainty aligns with true uncertainty. But that problem isn't tested here.

not sure why a "lambda" term comes in suddenly in (6). Also, is (5) correct? when the predicted and true labels are the same, shouldn't the score be the entropy of the predicted outputs, not 1 minus that? Especially as CAU is minimized eventually (13). This also leads to another point of confusion in Table 2, as the higher CAU is in bold there. I thought the objective was to minimize CAU? (8) seems to have the same issue. Overall there seem to be quite a number of math typos.

Table 2 also shows a very significant dip in accuracy. While the other measures, especially AUPR-Error shows a decent increase, the significant drop in accuracy is still something that ideally should be avoided.

I question the premise of the experiment in 4.1.1. The authors primarily use the correlation of the uncertainty measure with the degree of dropout rate, citing that higher dropouts must yield somehow models that are better in "uncertainty quality". It is not clear what that actually refers to. The metrics which are chosen here for comparison (AUROC, AUPR-Error etc.) are primarily for the failure prediction task, so I'm assuming the objective is failure prediction. The corresponding assumption that the paper makes here is that failure prediction/misclassification detection "must" be easier for networks with larger dropout. Why is this true? There needs to be more evidence for the same. Also, isn't the thing to do here is to compare usual UQ approaches (like CE loss) with CAU across the various failure detection metrics, and see if the AUROC, AUPR etc. is better for CAU than other standard measures. In that case that will cement the fact that CAU is better at identifying missclassifications, and thus UQ.


Also, increasing dropout isn't the only way a model can increasingly become uncertain in its predictions. In fact, dropout aligns with the motivation of the proposed metric, as dropout also should yield more spread out confidence as it is increased, agreeing with the motivation of CAU which also imposes lower confidence (higher entropy) overall when the model gets worse/more uncertain in its prediction. However, there can be data driven uncertainty as well, which occurs because more corrupted data points are introduced in the training. Here the model still stays "confidently wrong" in its prediction of the corrupted data, so the flattening of entropy doesn't necessarily occur.

How is $\lambda$ chosen, which is one of the parameters when estimating the CAU? I did not find it in the paper.

A more in-depth theoretical discussion on the proposed metric and how it relates to binary decision theory would be required to see the benefits of the proposed measure.

Lastly, I believe the experiments have to be expanded to better make the case for using CAU, both as a measure for UQ and as a regularizer. More architectures and larger datasets will need to be tested.

**Questions:**

Please see weaknesses.

---

### Meta-Review · Area_Chair_Z6hZ · 2025-12-25

**Summary:**

The reviewers raised numerous concerns and unanimously recommended rejection. The authors have not provided any rebuttal. The paper is not suitable for the conference in its current form.

**Reviewer Concerns:**

There was no rebuttal.

**Reviewer Scores:**

There was no rebuttal.

---

### Decision · Program_Chairs · 2026-01-26

Reject